# Local Adaptation and Response of *Platycladus orientalis* (L.) Franco Populations to Climate Change

**Xian-Ge Hu [1,2], Jian-Feng Mao [1,]\***, **Yousry A. El-Kassaby [2], Kai-Hua Jia [1], Si-Qian Jiao [1], Shan-Shan Zhou [1], Yue Li [1], Nicholas C. Coops [3] and Tongli Wang [2,]\***

1   Beijing Advanced Innovation Center for Tree Breeding by Molecular Design, National Engineering Laboratory for Tree Breeding, Key Laboratory of Genetics and Breeding in Forest Trees and Ornamental Plants, Ministry of Education, College of Biological Sciences and Technology, Beijing Forestry University, Beijing 100083, China
2   Department of Forest and Conservation Sciences, Faculty of Forestry, The University of British Columbia, 2424 Main Mall, Vancouver, BC V6T 1Z4, Canada
3   Department of Forest Resources Management, Faculty of Forestry, The University of British Columbia, 2424 Main Mall, Vancouver, BC V6T 1Z4, Canada
\*   Correspondence: jianfeng.mao@bjfu.edu.cn (J.-F.M.); tongli.wang@ubc.ca (T.W.);
    Tel.: +86-13366181735 (J.-F.M.); +1-604-822-1845 (T.W.)

**Abstract:** Knowledge about the local adaptation and response of forest tree populations to the climate is important for assessing the impact of climate change and developing adaptive genetic resource management strategies. However, such information is not available for most plant species. Here, based on 69 provenances tested at 19 common garden experimental sites, we developed a universal response function (URF) for tree height at seven years of age for the important and wide-spread native Chinese tree species *Platycladus orientalis* (L.) Franco. URF was recently used to predict the potential growth response of a population originating from any climate and growing in any climate conditions. The developed model integrated both genetic and environmental effects, and explained 55% of the total variation in tree height observed among provenances and test sites in China. We found that local provenances performed better than non-local counterparts in habitats located in central, eastern, and southwestern China, showing the evidence of local adaptation as compared to other regions. In contrast, non-local provenances outperformed local ones in peripheral areas in northern and northwestern China, suggesting an adaptational lag in these areas. Future projections suggest that the suitable habitat areas of *P. orientalis* would expand by 15%–39% and shift northward by 0.8–3 degrees in latitude; however, the projected tree height of this species would decline by 4%–8% if local provenances were used. If optimal provenances were used, tree height growth could be improved by 13%–15%, along with 59%–71% suitable habitat expansion. Thus, assisted migration with properly selected seed sources would be effective in avoiding maladaptation in new plantations under a changing climate for *P. orientalis*.

**Keywords:** provenance test; *Platycladus orientalis*; climate change; universal response function; assisted migration; adaptation

## 1. Introduction

Scientific evidence indicates that the global temperature will continue to rise unless drastic measures are taken to reduce persistent carbon emissions [1]. Even the most optimistic climate change scenarios also project the average temperature of Earth's surface to increase by 1.4 °C by 2100 [2].



Climate warming affects many aspects of plant growth, such as the range shifts of the species suitable habitats (i.e., either poleward or upward in elevation) and changes in morphology and phenology, affecting breeding and regeneration [3]. Several studies have provided compelling evidence that the phenology and adaptive traits of tree populations will be affected by climatic change [4–7]. For example, the vulnerability of spruce and beech would increase in the period to 2100, and their populations could become locally maladapted and decline in fitness [5].

Tree populations are generally adapted to their local habitats through long-term natural selection, migration and phenotypic plasticity [8]. It is widely agreed that the natural migration of forest trees is lagging far behind the current rapid rate of climate change [9–12]. Therefore, assessment of the potential growth response of forest trees to climate change and the selection of optimal provenances are critical to improving the survival rate of tree populations facing climate change [13]. Since the species distribution model (SDM) was first used to explore the distributional limits of plant species in 1980s [14,15], it has been widely used to predict potential changes in species distributions under climate change scenarios [16–19]. However, SDMs treat tree species as single homogeneous entities and cannot predict growth rates; thus, among-population variation within tree species could not be considered in developing forest adaptive strategies through seed transfer and assisted migration.

Population transfer and response functions have previously been used to estimate the genetic and environmental effects on the potential growth of forest trees. A transfer function relates the performance of different populations to the climatic transfer distances between a given plant site and its origin source [20–24], reflecting the local adaptation to climate (genetic effect). A population response function reflects the environmental effects of site climate variables on the performance of a given population [23,25,26]. In order to integrate both the genetic and the environmental effects into a single model to better predict the influence of climate on growth performance, Wang et al. [27] developed a "universal response function" (URF), which can predict the potential growth response of a population originating from any climate and growing in any climate conditions. This approach has been applied in several species in North America [28] and Europe [29,30]. Additionally, URF can also be applied to identify the optimal provenance for each planting site [27] and predict the performance of populations using both local and optimal provenances. In terms of an observed trait, the performance differences between local and optimal provenances can then be used to assess the local adaptation driven by climate. Accordingly, the URF approach was adopted in the present study.

*Platycladus orientalis* (L.) Franco (Genus: *Platycladus* Spach) is a wide-spread species native to China, and has a long history of domestication. The species is well known for its resistance to harsh environmental conditions such as cold, drought, high temperature, salt, and barren land [31–34]. Despite its slow-growth rate, this species is one of the important afforesting tree species in northern China, especially for establishing plantations on barren mountains. For example, about 26% of the newly planted plantations are *P. orientalis* around Beijing [35]. However, in China, annual mean temperature increased by 0.78 °C during the 1906–2005 period [36], and is projected to increase by 1.7–2.0 °C over the next 40–60 years [37], which is higher than the world average. Therefore, detailed information that helps to choose the optimum provenances for afforestation and reforestation is important to avoid maladaptation under a rapidly changing climate. To improve the adaptability and productivity of forests, the forestry community in China has initiated seed certification and established seed zone delimitation based on large-scale provenance tests for this species in 1980s. The impacts of climate change at both species and seed zone levels have been previously assessed using climate niche models, and the results indicated that *P. orientalis* matching current climates may grow in sub-optimal conditions in the future [18,19]. However, the local adaptation and response to climate change in growth performance at the population level remain unexplored.

The main goal of this study was to understand the local adaptation and response of *P. orientalis* to climate at the population level and to provide a scientific basis for the development of adaptive strategies. The specific objectives were to: (1) develop a UFR model to predict the growth response of populations to climate; (2) examine the spatial patterns of local adaptation; (3) assess the potential

impact of future climate change on growth and habitat shift, and (4) provide recommendations for optimizing *P. orientalis* provenances for afforestation and reforestation in China.

## 2. Materials and Methods

### 2.1. Plant Materials

The present study is based on tree height data collected from a provenance trial series established by an industry cooperative in 1980s (Table 1). Briefly, the provenance trial series represent a range-wide collection of 112 *P. orientalis* populations (provenances) tested at 32 test sites in China (Figure 1), which cover the main habitat of *P. orientalis* in China [18,19] (https://www.conifers.org/cu/Platycladus.php). Only native (i.e., autochthonous) populations were sampled, and seeds were collected from three trees to represent each of the 112 populations. Where possible, the seed-tree donors were spaced apart by at least 100 m to reduce the likelihood of including relatives. The locations of the 32 test sites were chosen to maximize the representation of the large environmental gradients of the entire region suitable for *P. orientalis* in China. The number of provenances tested at each site varied between 3 and 70 with a majority above 20. At each test site, the tested provenances (include local provenance) were planted with one-year-old seedlings and arranged using a randomized complete block design with three replications in 75-tree-row plot at $1 \times 2$ m spacing. In the provenance trial series of *P. orientalis*, the collected seeds were planted in spring at 1983, 1984 and 1988, and the same cultivation practices were applied among different sites. In this study, most of the tested provenances were planted in 1983. Trees were measured for height after 1–7 growing seasons in the field. We used the provenance mean of individual trees from each test site to represent the growth performance of each provenance at each test site.

**Table 1.** List of test sites and studies reporting tree height at various ages (after plantation) on tree height for *Platycladus orientalis* (L.) Franco provenance trials.

| No. | Test Site | Latitude | Longitude | Elevation | No. of pop [1] | Ages of obs [2] | | Used in URF | Source |
|-----|-----------|----------|-----------|-----------|----------------|-----------------|-----------|-------------|--------|
| | | | | | | Provenance | Trial Mean | | |
| 1 | Guiyang | 25.93 | 106.55 | 957 | 33 | 1 | | | [38] |
| 2 | Nanping | 26.66 | 118.12 | 739 | 23 | 2 | 6 | Yes | [32,39] |
| 3 | Cili | 29.47 | 111.05 | 761 | 23 | 2 | 6 | Yes | [32,39] |
| 4 | Chongqing | 30 | 107.04 | 808 | 27 | 5 | | | [31] |
| 5 | Deyang | 31.68 | 104.5 | 988 | 16 | 1–2 | | | [40] |
| 6 | Queshan | 32.79 | 114.01 | 155 | 23 | 2 | 6 | Yes | [32,39] |
| 7 | Xuzhou | 33.73 | 117.78 | 23 | 20 | 1–7 | 7 | Yes | [41] |
| 8 | Liangdang | 33.99 | 106.25 | 1193 | 21 | 3 | 7 | Yes | [32,42] |
| 9 | Luonan | 34 | 110.35 | 1296 | 7 | 1–2 | 7 | Yes | [32,43] |
| 10 | Jiaxian | 34.1 | 113.18 | 642 | 70 | 1–2, 19 | | | [33,44] |
| 11 | Dengfeng | 34.35 | 113.07 | 334 | 14 | 2 | 7 | Yes | [32,39] |
| 12 | Chunhua | 34.79 | 108.54 | 1025 | 15 | 1–2 | 7 | Yes | [33,43] |
| 13 | Jiyuan | 34.88 | 112.95 | 110 | 25 | 1–2 | | | [45] |
| 14 | Zaozhuang | 34.99 | 117.71 | 366 | 18 | 2 | 6 | Yes | [32,46] |
| 15 | Zhengning | 35.39 | 108.49 | 1260 | 21 | 3 | | | [42] |
| 16 | Jishan | 35.77 | 110.99 | 1597 | 23 | 7 | 7 | Yes | [47] |
| 17 | Yanan | 36.06 | 109.26 | 971 | 28 | 1–2 | | | [48] |
| 18 | Pingyin | 36.08 | 116.33 | 297 | 21 | 1–7 | 7 | Yes | [49] |
| 19 | Lanzhou | 36.12 | 103.71 | 1551 | 21 | 3 | 7 | Yes | [32,50] |
| 20 | Yidu | 36.41 | 118.27 | 575 | 19 | 2 | | | [46] |
| 21 | Yuanshan | 36.49 | 117.86 | 250 | 37 | 23 | 7 | Yes | [32,51] |
| 22 | Taiyuan | 37.76 | 112.3 | 1799 | 51 | 1 | | | [52] |
| 23 | Yulin | 39.03 | 111.08 | 868 | 14 | 1 | | | [53] |
| 24 | Mentougou | 39.98 | 116.05 | 157 | 29 | 1 | 6 | Yes | [54] |
| 25 | Haidian | 40.01 | 116.34 | 51 | 26 | 1–3, 5–6 | 7 | Yes | [32,33] |
| 26 | Zunhua | 40.19 | 117.63 | 260 | 20 | 1–6 | 7 | Yes | [32,55] |
| 27 | Helin | 40.26 | 112.06 | 1262 | 27 | 1–2 | | | [56] |
| 28 | Datong | 40.35 | 113.58 | 1392 | 8 | 1–7 | 7 | | [57] |
| 29 | Miyun | 40.4 | 117.06 | 206 | 24 | 1 | | | [32,53] |
| 30 | Xingcheng | 40.64 | 120.77 | 43 | 16 | 1–3 | 7 | Yes | [32,50] |
| 31 | Zhuozi | 40.97 | 112.15 | 1437 | 3 | 1–2 | | | [56] |
| 32 | Lingyuan | 41.19 | 118.98 | 1209 | 10 | 2 | 7 | Yes | [32,39] |

[1] Number of the tested populations. [2] Ages of the observed tree height for individual provenances and trial mean.

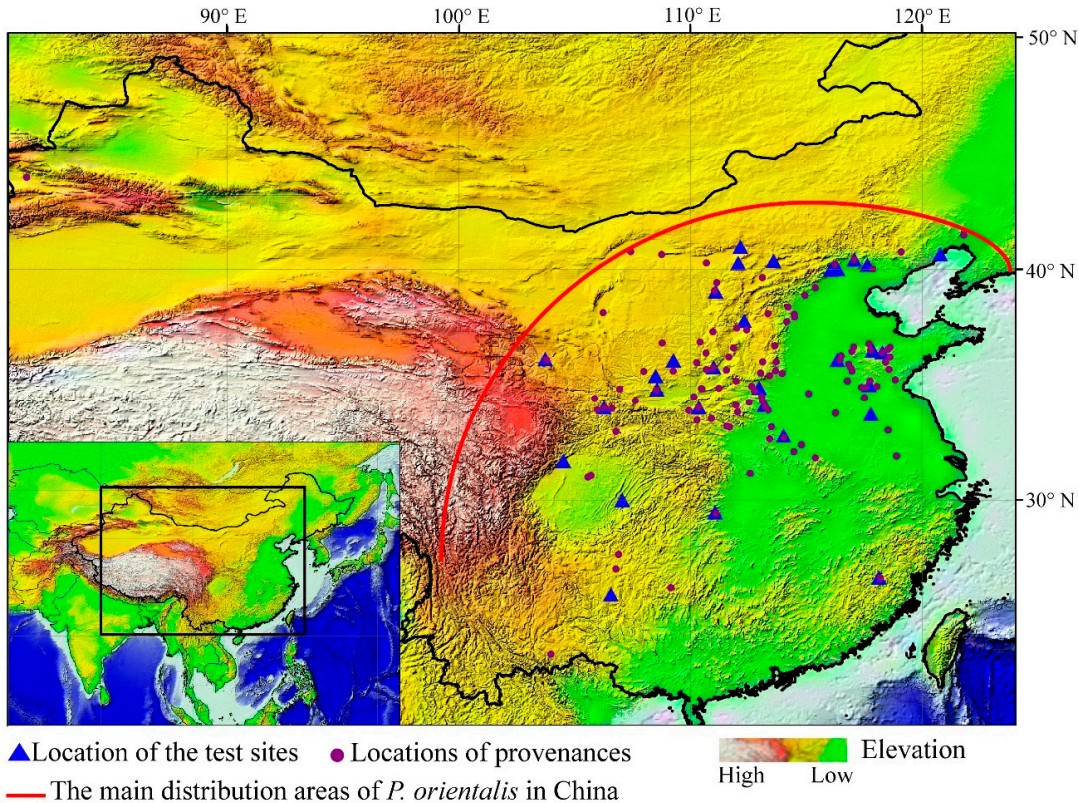

**Figure 1.** Distributions of the 32 test sites and the 112 provenances of the provenance trails. Test sites in blue triangles and provenances in violet circles.

In this study, we used tree height as the response variable, as it is known to be under strong selection (often considered as fitness proxy) and has a strong genetic component (i.e., high heritability) [34,58]. Previous studies have demonstrated that it's possible to model the relationships between tree height and age [59]. Additionally, the relationship of *P. orientalis* tree height in 1–7 ages have been analyzed in six provenance test sites (Lanzhou, Liangdang, Zunhua, Pingyin, Jishan and Haidian) and the results showed that tree height growth was closely related to ages, and the correlation coefficient grows with the decrease of age difference [32,33,42,47,49,50,55]. Therefore, as 7-year old height data were only available for 14 test sites, we utilized the height-age relationships to predict 7-year old height for the remaining 18 test sites.

### 2.2. 7-Year Tree Height Estimation Method

To estimate 7-year height of each provenance (*i*) at each test site (*j*), we first estimated the general mean over all sites $\left(\overline{HT}_{ij}\right)$, then the site means over provenances, and finally the provenance values at each test site at each age. The following regression formula was used to estimate the general mean:

$$HT_j = a + bX \tag{1}$$

where, $HT_j$ is the observed site mean height at the *j*th site; *X* is age of the observation; and *a* and *b* are the parameters to be estimated. We used the predicted site mean as the general mean at each age.

A site mean was estimated based on the relative deviation of the site from the general mean. The relative deviation of each observed site mean height from the general mean was calculated as:

$$R_j = \frac{HT_j - \overline{HT}_{ij}}{\overline{HT}_{ij}} \tag{2}$$

where, $R_j$ is the relative deviation of the $j$th site from the general mean $\overline{HT}_{ij}$. The $R_j$ available at the oldest age (i.e., closest to seven years old) was used to adjust the general mean height $\overline{HT}_{ij}$ at seven years old ($\overline{HT7}_{ij}$) to obtain site mean height ($HT7_j$) as:

$$HT7_j = \overline{HT7}_{ij} \times \left(1 + R_j\right) \tag{3}$$

where, $HT7_j$ is the predicted site mean HT7 of the $j$th site; $\overline{HT7}_{ij}$ is the estimated general mean tree height at seven years old.

Similarly, to estimate HT7 of each population at each site ($HT7_{ij}$), we first calculated the relative deviation of each population from the site mean HT ($HT_j$) at each age as:

$$R_{ij} = \frac{HT_{ij} - HT_j}{HT_j} \tag{4}$$

where, $R_{ij}$ is the relative deviation of the $i$th population at the $j$th test site from the $j$th test site mean.

The $R_{ij}$ available at the oldest age was used to adjust the $j$th site mean at seven years old ($HT7_j$) to obtain $HT7_{ij}$ as:

$$HT7_{ij} = HT7_j \times \left(1 + R_{ij}\right) \tag{5}$$

where $HT7_{ij}$ is the predicted height of the $i$th population at the $j$th test site.

Although the HT7 of tested populations at the 18 test sites could be estimated from observed height at ages 1–6, we considered the ages 1–2 to be too young, while ages 3–5 were covered by the age 6 with only one exception. Therefore, only HT7 estimated based on the observation at age 6 were included, which added five more test sites to the dataset and increased the total number of test sites from 14 to 19 for further analysis, which includes 69 tested provenances. Other test sites also contributed to the estimation of the height–age relationship and the general means.

## 2.3. Climate and Geographic Variables

ClimateAP [60] was used to generate climate data for the reference period 1961–1990 and future periods. ClimateAP is a standalone MS Windows software application that extracts and downscales gridded ($4 \times 4$ km) monthly climate data for the reference normal period (1961–1990) from PRISM [61] and WorldClim [62] to scale-free point locations, rather than using the averages over a grid from other climate models. It generates >200 monthly, seasonal, and annual climate variables. The downscaling is achieved through a combination of bilinear interpolation and dynamic local elevational adjustment. ClimateAP also uses the scale-free data as a baseline to downscale historical and future climate variables for individual years and periods between 1901 and 2100. In the case of future climate data, two greenhouse gas representative concentration pathways (RCP 4.5 and 8.5) and 15 general circulation models are included in ClimateAP. In this study, climate variables were obtained for all test sites and provenances for the reference period using ClimateAP of version 2.11 for developing the URF model.

We selected 14 temperature and moisture-related climate variables (Table 2) of seed-source locations and test sites for the decadal mean of the 1980s for the analysis in this study. The year-to-year variation in climate associated with the difference in planting years was neglected as it was small relative to the magnitude of spatial climate variation and climate change. These variables were chosen based on: (1) the availability of past, present, and future climate data for all seed-source locations and test sites; (2) similarity to variables used in other genecological studies and frequently used in climate niche modeling of *P. orientalis* [18,19] to facilitate comparisons, and (3) biological relevance to plant growth, for example, the eight selected climate variables that closely relate to temperature were of significant biological relevance, because temperature has proven to be a very relevant climate factor for the growth of *P. orientalis* [18,19]. The 14 climate variables (Table 2) and the extension of a climate variable name (e.g., MAT) "_s" or "_p" were appended to each variable to denote test sites or provenances, respectively. Geographic variables were also used to reflect among-population

variation that is driven by non-climatic factors such as gene flow, demographic history, and colonization events. They included latitude (LAT), longitude (LONG), elevation (EL), and their transformations and interactions (LAT$^2$, LONG$^2$, EL$^2$, LAT × LONG, LAT × EL, LONG × EL, (LAT × LONG)$^2$, (LAT × EL)$^2$, (LONG × EL)$^2$, LAT × LONG$^2$, LAT × EL$^2$, LAT$^2$ × LONG, LONG × EL$^2$, LAT$^2$ × EL, LONG$^2$ × EL).

**Table 2.** Climate variables that were used to develop a universal response function for tree height growth of *P. orientalis*.

| Code | Variable Name |
|------|---------------|
| MAT (°C) | Mean annual temperature |
| MWMT (°C) | Mean warmest month temperature |
| MCMT (°C) | Mean coldest month temperature |
| TD | Continentally, temperature difference between MWMT and MCMT |
| MAP (mm) | Mean annual precipitation |
| AHM | Annual heat-moisture index (MAT + 10)/(MAP/1000)) |
| DD < 0 (°C) | Degree-days below 0 °C |
| DD > 5 (°C) | Degree-days above 5 °C |
| NFFD (day) | Number of frost-free days |
| EMT (°C) | Extreme minimum temperature over a 30-year period |
| EXT (°C) | Extreme maximum temperature over a 30-year period |
| PAS (mm) | Precipitation as snow |
| Eref | Hargreaves reference evaporation |
| CMD | Climatic moisture deficit |

In order to predict the geographical distribution of populations performance for current climatic conditions and to project for future climates, gridded climate data for all the selected climate variables were generated at spatial resolution of 4 × 4 km for three 30-year climate normal periods: 1961–1990 (reference period), and two future normal periods 2041–2070 and 2070–2100 [60]. These periods are hereafter referred to as: 1970s, 2050s, and 2080s, respectively. We chose two greenhouse gas representative concentration pathways (RCP 4.5 and 8.5) and the ensemble of 15 general circulation models (GCMs) included in ClimateAP for the two future periods.

### 2.4. Universal Response Function (URF)

Prior to the development of an URF, we conducted an analysis of variance (ANOVA) on HT7 to determine if significant variation exists among test sites and among populations. ANOVA is a useful statistical model simultaneously testing between-mean differences in more than two conditions. Additionally, linear mixed models have been developed to take into account the variation among test sites and among populations, because it allows incomplete and unbalanced data to be used, continuous and categorical predictors to be combined, and information loss due to averaging over observations or participants to be avoided [63] which is ideal for our data from an unbalanced design of provenance tests.

A linear mixed-effects model (lmer function in the "lme4" package) [64] was used:

$$Y_{ij} = u + P_i + S_j + e_{ij} \qquad (6)$$

where $Y_{ij}$ is the observation of the *i*th population at the *j*th site; $P_i$ is the fixed effect of *i*th population; $S_j$ is the fixed effect of *j*th site; and $e_{ij}$ is the random error. HT7 met the normal distribution and homogeneity of variance assumptions, thus no data transformation was implemented. The interaction between the site and population was not considered, as there was only one observation for each provenance at each test site.

The URF was developed following Wang et al. [27] with the following model:

$$Y_{ij} = b_0 + b_1 X_{1i} + b_2 X_{1i}^2 + b_3 X_{2j} + b_4 X_{2j}^2 + b_5 X_{1i} X_{2j} + b_6 X_{3j} + b_7 X_{3j}^2 + e_{ij} \qquad (7)$$

where $Y_{ij}$ is the observation of the *i*th population at the *j*th site; *b* denotes are the intercept and regression coefficients; $X_{1i}$ and $X_{2i}$ are one or more climatic variables for the test site and the provenance, respectively; $X_{3j}$ is one or more geographic variables for the *i*th provenance; $X_{1i}X_{2j}$ is the interaction(s) between $X_{1i}$ and $X_{2i}$; and $e_{ij}$ is the residual [27]. We used HT7 as the dependent variable ($Y_{ij}$).

Multiple regression models were constructed using different variable combinations with a maximum of three climate variables for site and with an additional geographic variable for provenance [27]. The best model was selected based on the highest proportion of variance explained by partial R$^2$ and lowest Akaike information criterion (AIC) value. In addition, the stepwise selection method was applied to filter out non-significant terms in the chosen regression model, and the condition number kappa (κ) of the chosen variables are calculated to diagnose the presence of multicollinearity.

In this study, all analyses were performed in the statistical computing environment R (v3.5.3; [65]). Spatial calculations for the distribution of *P. orientalis* in China, i.e., spatial modeling of population HT7 and estimating the growth performance of local and optimum populations under a future climate, were conducted using the R packages "raster", "maptools", and "rgdal".

## 2.5. Applications of URF

The URF can be used to predict the spatial distribution of the growth performances of local (i.e., provenances from the climate conditions that are the same as the planting site) or optimum provenances (i.e., the best performing provenance for the planting sites). We first estimated the growth of local provenances in the current climate. In this case, the URF was simplified into a population response function as the variables for the provenance were treated as constants, the climate variables for each planting site were equal to that of provenances in the URF (i.e., MAT_p = MAT_s and MWMT_p = MWMT_s). Second, we used URF to predict the potential future spatial distribution of the growth with local population under two future climate scenarios/periods (RCP4.5-2050, RCP4.5-2070, RCP8.5-2050, and RCP8.5-2070). In this case, the URF used current local climate variables for provenance and future climate values for test site. In total, 1,279,335 grid points (covering whole China) were predicted for each period, then the suitable area (with a predicted HT7) were calculated and the percent change in area (relative to the suitable area under current climate scenario and local provenances) were predicted under the different climate scenarios. Considering the maximum and minimum HT7 is 0.4 and 3.68 m in the provenance tests, we limit the predictive range of tree height between 0 and 4 m.

In order to identify the optimum population for each planting site, we followed the same procedure described in Wang et al. [27] by: (1) taking the first-order partial derivatives of the URF with respect to climatic and geographic variables of provenance, respectively; (2) setting these partial derivatives to zero, and (3) solving the equations to obtain the values of the provenance climate variable expected to provide greatest growth at the site. Based on the nature of the response of each climate variable—only one optimal (or the maximum) value is possible—so that we did not need to take the second derivative. After applying these optimum values to the URF, HT7 of each planting sites (grid point locations) with the optimum population (maximum height) was estimated for the reference (1970s) and the two future periods (2050s and 2080s).

## 3. Results

### 3.1. Estimated 7-Year Tree Height

The relationship between observed site mean height ($HT_j$) and the age of each observation across all the 32 test sites was strong ($R^2 = 0.63$ and $p < 0.01$). The estimated general mean height $\left(\overline{HT_{ij}}\right)$ at each age was the expected mean on the trend line as shown in Figure 2a. The relationship was stronger at each individual site for all the 14 sites with observations at multiple ages (Figure 2a). The estimated HT7 for the 18 test sites without observations at age 7 showed similar trends, suggesting that using the sites' mean HT7 was a reasonable approach (Figure 2b).

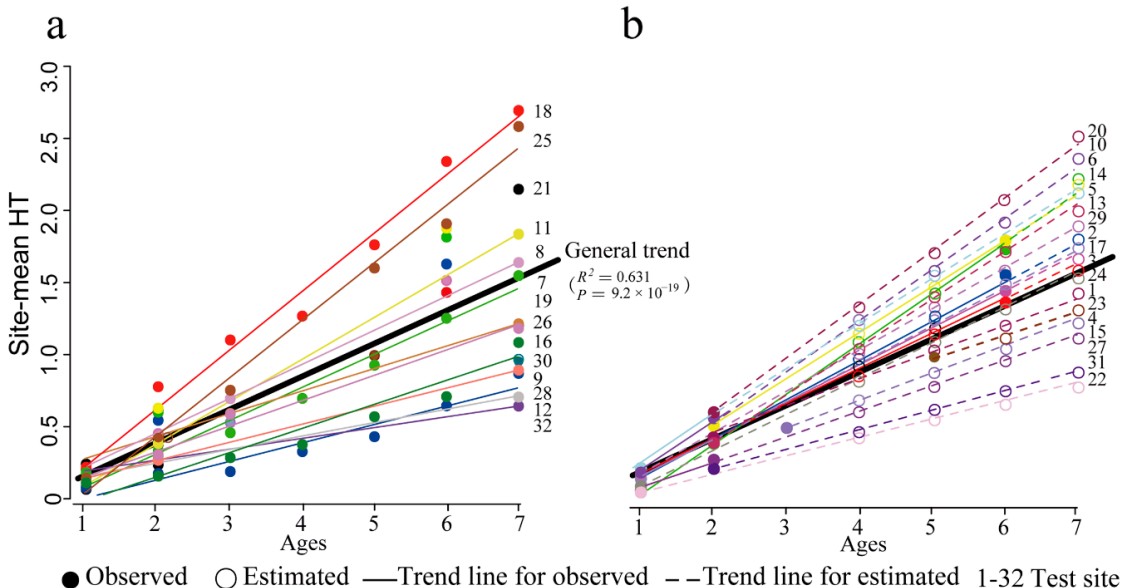

**Figure 2.** The linear relationship between the site-mean tree height (HT) and ages for test sites with observations at age 7 (**a**) and test sites without observations at age 7 (**b**).

The trend of height growth was maintained after adding the estimated tree height to each tested population, as illustrated in Figure 3 for the test sites of Haidian and Xingcheng, suggesting reasonable estimations of HT7 for each tested population. In total, HT7 was obtained for 69 tested provenances at the test site where the tree height measured at age six and seven.

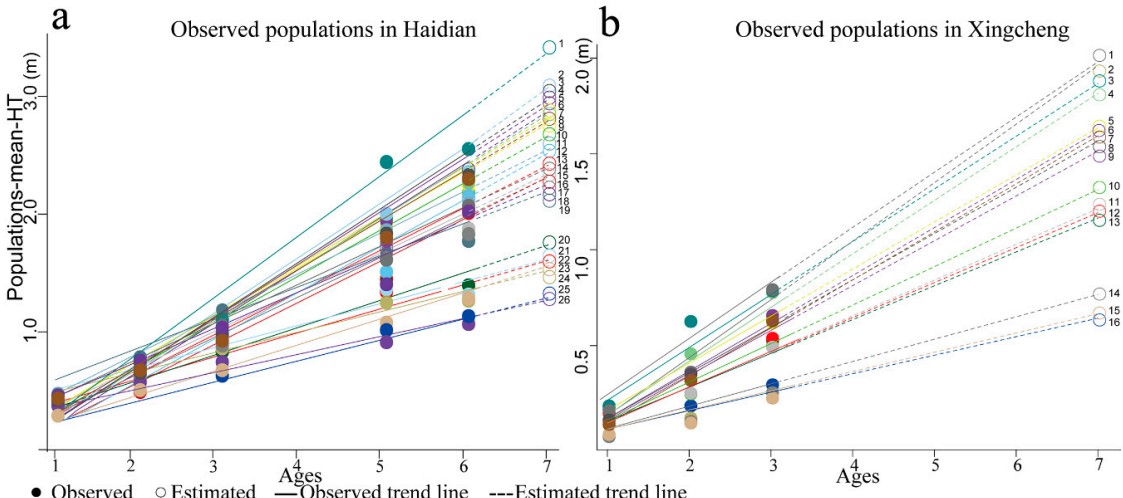

**Figure 3.** The estimated growth trend of the tested populations in test site of Haidian and Xingcheng. Observed populations in Haidian: 1 taian; 2 yixian; 3 pingyin; 4 shilou; 5 huolu; 6 licheng; 7 huixian; 8 zunhua; 9 dengfeng; 10 changping; 11 miyun; 12 qingzhou; 13 changzhi; 14 xuzhou; 15 weixian; 16 linru; 17 heshui; 18 yongcheng; 19 queshan; 20 luonan; 21 nanping; 22 chunhua; 23 shehong; 24 zhongyang; 25 baotou; 26 wenshui (**a**). Observed populations in Xingcheng: 1 xuzhou; 2 shehong; 3 taian; 4 licheng; 5 zunhua; 6 huolou; 7 dengfeng; 8 yixian; 9 shilou; 10 changqing; 11 lanzhou; 12 changzhi; 13 luonan; 14 weixian; 15 zhongyang; 16 baotou (**b**).

*3.2. Universal Response Function*

In this study, we adopted the URF approach to show how the observed HT7 differed among populations and test sites (Table 3), which showed that significant variation exists. After comparisons of the performance for different variable combinations (from step 1 to 5 in Table S1) in URF, and screened

out non-significant terms by stepwise selection method, the best model (AIC = −543.88) was chosen, which includes three climatic variables (five combinations) for site (DD5_s and DD5_s$^2$, MAP_s and MAP_s$^2$, CMD_s), two climatic variables for provenances (MAT_p and MAT_p$^2$), and three interactions terms between site and provenance climatic variables (DD5_s × MAP_s, DD5_s × CMD_s, MAP_s × CMD_s), and accounted for 55% of the total variation in HT7 (Table 4). Additionally, the calculated Kappa (39.08) lower than 100 showed that there is a low level of multicollinearity between the chosen independent variables.

**Table 3.** Analysis of variance of tree height.

| Source of Variation | Degree of Freedom | Sum of Squares | Mean Squares | F Value | Pr (>F) Value |
|---|---|---|---|---|---|
| Site | 18 | 121.6 | 6.756 | 64.989 | <2 × 10$^{-16}$ |
| Population | 68 | 29.45 | 0.433 | 4.167 | <2 × 10$^{-17}$ |
| Residuals | 283 | 29.42 | 1.104 | | |

**Table 4.** Results of multiple regression analysis predicting *P. orientalis* height using site and provenance climate variables in a universal response function.

| | Estimate | Std. Error | T Value | Pr (>\|t\|) | Significance |
|---|---|---|---|---|---|
| (Intercept) | −8.31 | 1.40 | −5.95 | 6.33 × 10$^{-9}$ | *** |
| MAP_s | −6.71 × 10$^{-3}$ | 2.11 × 10$^{-3}$ | −3.18 | 1.59 × 10$^{-3}$ | ** |
| DD5_s | 5.11 × 10$^{-3}$ | 8.14 × 10$^{-4}$ | 6.27 | 9.87 × 10$^{-10}$ | *** |
| CMD_s | 9.25 × 10$^{-3}$ | 3.73 × 10$^{-3}$ | 2.48 | 1.35 × 10$^{-2}$ | * |
| MAT_p | 4.31 × 10$^{-1}$ | 6.95 × 10$^{-2}$ | 6.20 | 1.54 × 10$^{-9}$ | *** |
| MAP_s$^2$ | 1.99 × 10$^{-5}$ | 4.06 × 10$^{-6}$ | 4.90 | 1.41 × 10$^{-6}$ | *** |
| DD5_s$^2$ | 1.18 × 10$^{-6}$ | 3.86 × 10$^{-7}$ | 3.04 | 2.47 × 10$^{-3}$ | ** |
| MAT_p$^2$ | −1.29 × 10$^{-2}$ | 2.63 × 10$^{-3}$ | −4.9 | 1.45 × 10$^{-6}$ | *** |
| MAP_s × DD5_s | −1.11 × 10$^{-5}$ | 2.55 × 10$^{-6}$ | −4.33 | 1.87 × 10$^{-5}$ | *** |
| MAP_s × CMD_s | 2.59 × 10$^{-5}$ | 4.82 × 10$^{-6}$ | 5.36 | 1.43 × 10$^{-7}$ | *** |
| DD5_s × CMD_s | −1.02 × 10$^{-5}$ | 2.24 × 10$^{-6}$ | −4.54 | 7.68 × 10$^{-6}$ | *** |

Abbreviations are: MAT, mean annual temperature; DD5, degree-days above 5°; CMD, climatic moisture deficit; MAP, mean annual precipitation. The extension "_s" or "_p" was appended to each variable to denote test sites or provenances, respectively. * represent significant, ** represent very significant, *** represent extremely significant.

In order to illustrate the universal response function graphically (3-D graph), we used a simplified model with two independent variables, and their interactions. Thus, the simplified function involving the best combination of the two climatic variables DD5_s and MAT_p, explaining 49% of HT7 total variation (Figure 4). The response surface (Figure 4) describes the majority of phenotypic variation among populations in growth response to site climate, and thus is a good representative of the observed trends in the data.

### 3.3. Spatial Variation in Growth Performances

The predictions of HT7 based on the URF (Table 4) using the local provenances showed that there was a wide range of areas suitable for *P. orientalis* in China under the current climate (about 3.74 × 10$^6$ km$^2$) (Figure 5a; Table 5). Regions showing better performances were in central, eastern, and south-eastern China. In contrast, the performance was relatively poor in the northern, northwestern, and northeastern regions. Across the entire suitable habitats, the mean HT7 was about 2.0 m.

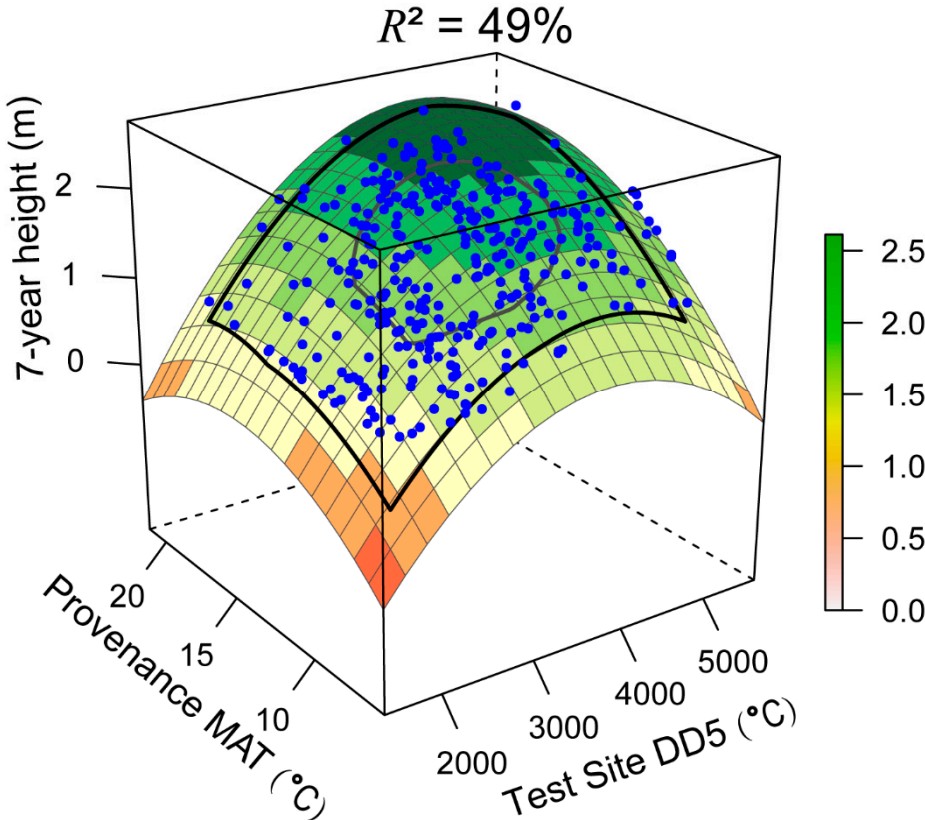

**Figure 4.** Observed heights of populations in *P. orientalis* provenance tests and modeled universal response function in which height is predicted as a joint function of test site DD5 and provenance MAT.

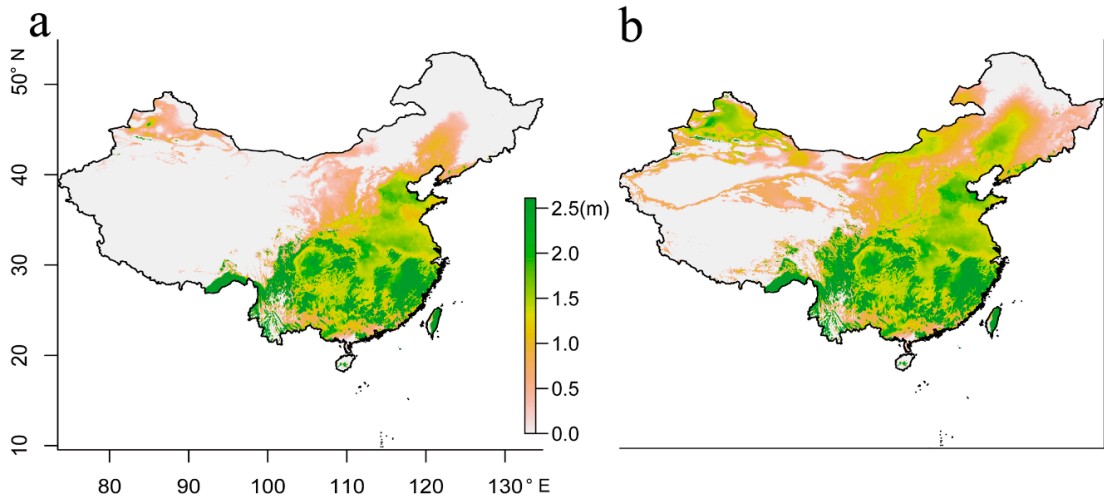

**Figure 5.** Spatial predictions of the universal response function for height at age 7 for *P. orientalis* populations under the current climate using local (**a**) and optimal provenances (**b**).

When optimal provenances were used at all planting sites, HT7 predicted performance showed substantial improvement amounting to a 17% (2.4 m) increase as compared to those using local provenances (Figure 5b). Additionally, this could lead to an increase in the suitable habitat areas by 67% (approximately $6.26 \times 10^6$ km$^2$) (Table 5). This improvement was predicted to be mainly distributed in north and northwest of China, which is dominated by arid and semi-arid climates. In contrast, no improvement was noticeable in central distribution areas of this species.

**Table 5.** Changes in areas suitable for *P. orientalis* under different climate scenarios and periods. Areas with predicted HT7 above zero are considered as suitable habitat.

| Climate Change Scenario | Area ($\times 10^6$ km$^2$) | | Change in Area (%) [1] | |
|---|---|---|---|---|
| | Local | Optimal | Local | Optimal |
| Current | 3.74 | 6.26 | | 67.40 |
| RCP4.5-2050 | 4.56 | 6.40 | 22.01 | 71.28 |
| RCP4.5-2080 | 4.77 | 6.43 | 27.50 | 71.95 |
| RCP8.5-2050 | 4.88 | 6.47 | 30.44 | 72.95 |
| RCP8.5-2080 | 5.45 | 6.70 | 45.83 | 79.32 |

[1] The percent change is relative to the area predicted under current climate scenario using local provenances.

### 3.4. Projections for Future Climates

The overall performance of *P. orientalis* with local provenances were projected to have a moderate improvement in 2050s and 2080s following the moderate greenhouse gas emission of scenario (RCP4.5). This trend was especially evident at higher latitudes (Figure 6a), with increase of the suitable habitat area by 22% and 28% for 2050 and 2080, respectively (Table 5). However, mean HT7 for the entire suitable habitats would decrease by 3.8% and 3.56% in 2050s and 2080s relative to the current conditions. The predicted HT7 with local provenances showed that the suitable area for *P. orientalis* would move northward, and the performance of southern provenances would decline (Figure 6a). However, when optimal provenances were used at all planting sites, the area of suitable habitats would increase by 71.28% and 71.85% in the 2050s and 2080s, respectively (Figure 6a and Table 5), the mean HT7 of the entire habitats would be beyond the current level (Figure 6a). The increases in the area of suitable habitat were mainly concentrated in northern and northwestern China (Figure 6a), where the arid and semi-arid climates mean that local populations grow extremely slowly.

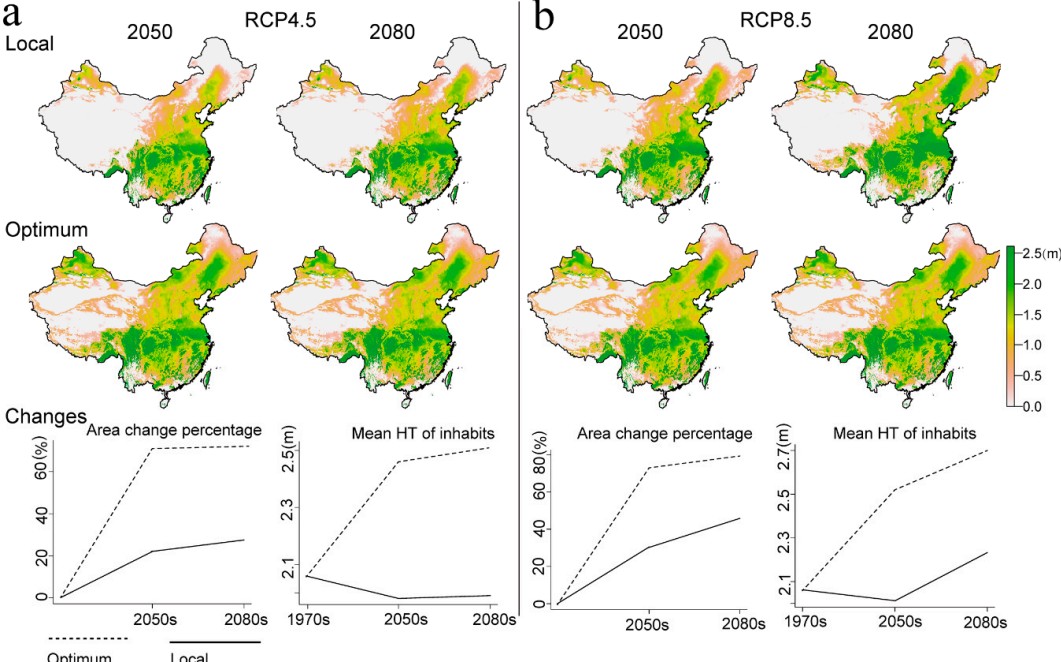

**Figure 6.** Maps of projected distribution of HT7 using local and optimal provenances, and changes in areas of suitable habitats in the future climates under RCP 4.5 (**a**) and RCP 8.5 (**b**).

If the concentration of greenhouse gas rises sharply in the future (RCP8.5), the impact of climate change on areas of suitable habitats and the performance would also follow the same trend as under that of the RCP 4.5 scenario, however the magnitude of the impact would be more drastic (Figure 6b).

For example, the predicted areas with local populations would increase by 31% and 46% in 2050s and 2080s, respectively, which are greater than that under RCP 4.5 scenario (Figure 6b; Table 5). Additionally, the area of suitable habitats is also projected to expand northward under RCP8.5, but the shift was farther north than that under the RCP4.5 scenario.

## 4. Discussion

To study the growth responses of *P. orientalis* populations to climate change and to understand the status of local adaptation, we developed a universal response function (URF) that incorporates both the genetic and environmental effects of climate into a single model. Using the URF model, we not only identified the most important climatic variables driving genetic variation among provenances (i.e., MAT) and environmental variation among test sites (i.e., DD5, MAP and CMD), but also predicted tree height growth performances using local and optimal provenances and revealed the spatial patterns of growth potential of this species over the country. If optimal provenances were used, substantially improved growth would be achieved along with the expansion of areas climatically suitable for this species, providing a promising solution to avoid maladaptation for new plantations under a changing climate.

### 4.1. Model Performance

Of the studied 32 provenance trials, 18 did not have 7-year-old tree height (HT7) data available. We developed a method to estimate the HT7 for these sites based on the height–age of tree height growth, which is found to be strong in previous studies for this species [32,47,50,66,67]. In the present study, we found that the age-to-age relationship was highly significant at each of the 14 test sites with observations at multiple ages. Our estimated HT7 for these sites maintained such relationships as our method for HT7 estimation incorporated the general trend of growth with age (the use of general regression line), the variation among test sites at a given age (nearest to age 7) and the variation among populations at each site at a given age (nearest to age 7). The observed significant height-age relationship of this species along with our developed robust method permitted estimating tree height as closely as the real-growth trend of different tested populations and provided a solid foundation for our model development. In this study, the condition number kappa (κ) showed that there was low level of multicollinearity in URF model. Besides, the variance inflation factor (VIF) also was analyzed to diagnose multicollinearity, but the result showed that there was still potential collinearity between the chosen variables. A possible explanation is that VIF diagnoses multicollinearity to a model in which all predictor variables are uncorrelated [68], but our chosen variables have a degree of correlation (because we focused on the top important variables to *P. orientalis*). Multicollinearity is often concerned in multiple regression for understanding the contribution of each predictors, but usually not a concern for the prediction power of the model, which is the main objective of our study. Therefore, considering the good performance of the condition number (κ) and that all of the variables are significant in the URF model, it's reasonable to use this model to predict the performance of *P. orientalis*.

The universal response function uses both provenance climate and site climate as predictors to represent the genetic and environmental effects of climate on tree height growth, respectively. A similar approach has been applied to facilitate the climate-smart regeneration of black spruce and white pine in Canada [28] and showed improvements in the model performance over the conventional response functions developed for lodgepole pine [26,27]. The models built for Douglas fir, based on observations in Austria and part of Germany, can predict the performance of this species in Europe [30] and the climate niche of this species in North America [29], demonstrating a robust feature of the URF models. Our URF explained 55% of the total variation in HT7. This value is within the range of the above-mentioned studies, but it is greater than that for general transfer functions. For example, they explained on average 30% of the total variation in height for 11 species in North America [20,69]. The random error of URF could result from population sampling effect, site-to-site edaphic (non-climatic environmental) variation [70], or biotic factors (i.e., insect, disease,

or animal damage). Considering these various sources of potential errors, our model performance was reasonable, allowing us to draw valuable conclusions from adaptation predictions across time, populations, and regions.

*4.2. Spatial Variation in Height Growth*

The importance of local adaptability in plants is widely recognized [71], and the mechanisms by which natural selection acts, over time to increase the frequency of genes relevant to local adaptability are largely understood [72]. For a wide-spread tree species, populations are often adapted to local climate conditions, resulting in spatial variation among populations in adaptive traits that often follow climatic gradients [73]. Populations growing under favorable climatic conditions may facilitate natural selection favoring fast growing individuals. In the present study, we found that such climatic conditions for *P. orientalis* exist in central and eastern China, where HT7 of this species is generally taller than that in other habitats with local provenances (Figure 5a). The central and eastern regions of China were also identified as origin centers or as major distribution areas of *P. orientalis* in previous studies [18,19,34]. Thus, local provenances also have the highest adaptability in these areas, where the general rule of "local is best" holds true [19,45,47,74].

On the other hand, we found that in some non-central areas, mainly distributed in northern and northwestern China (Figure 5), the performance of local provenances was much lower than that of the optimal ones (non-local provenances) (Figure 5b). One possible explanation is that natural selection for local adaptation is acting on other adaptive traits (i.e., body size, leaf shape) rather than height growth, although tree height is often considered as a good surrogate for fitness. Populations growing under suboptimal conditions may favor individuals with specific mechanisms to deal with the less favorable environmental conditions, such as resistance to drought or frost damage. Adaptational lag may be another explanation, which refers to a mismatch between genotypes and environments, caused by a relatively fast environmental change and a comparably slow evolutionary response [75]. It is reasonable to assume that populations in the central distribution areas have experienced a longer period of local adaptation resulting in less adaptational lags than those in the non-central distribution areas.

*4.3. Regional Variation in Adaptation to Future Climates*

Natural populations respond to global climate change by shifting their geographical distribution, phenotypic variation (such as body size, flowering time), and reproduction [76]. In the present study, we used tree height to predict the growth responses of *P. orientalis* to climate change. We not only projected that the habitat of *P. orientalis* would expand northward with climate warming, but also predicted the variation in tree height growth responses to climate among regions. In future climate scenarios of RCP4.5 and RCP8.5, tree height across the *P. orientalis* distribution areas would generally increase in northern China (Figure 6). This result suggests that the limitations of the marginal environmental conditions on the adaptability of this species would be reduced with climate warming, resulting in expected improvement in HT7 of *P. orientalis* in northern and northwestern regions of China. However, the projected height of southern populations showed a decreasing pattern with climate warming (Figure 6). It is evident that temperature changes will affect growth positively only within the species physiological and ecological tolerance limits [22]. The limit exceeded by temperature changes will obviously lead to losses in phenotype performance, and such effects were always expected at the southern limits of the *P. orientalis*' distribution area [18,19]. These findings are consistent with Hutchinson's theory of niches, that biology is limited by many environmental factors and each factor has a certain fitness threshold for a species to grow and reproduce normally, part of the niche with markedly suboptimal conditions near the boundary [77].

*4.4. The Risk to Local Populations Suffering from Maladaptation in Future Climates*

In this study, we projected a high maladaptation risk to local *P. orientalis* populations by the end of this century, especially for southern populations (Figure 6). This finding is consistent to the results

of our previous studies at both species level [19] and seed zone level [18]. However, projections in this study are also available for growth potential at the population level. We projected that the mean HT7 of the entire habitat would be reduced from 1970s to 2080s under both future climate scenarios of RCP4.5 and RCP8.5 (Figure 6). It is clear that the impact of climate change on tree growth is inevitable. Therefore, existing *P. orientalis* populations will likely suffer from maladaptation unless they can evolve or migrate quickly enough to keep pace with the rapidly changing climate.

The considerable within-population genetic variation in *P. orientalis* indicates a high potential for in situ evolution [39,44,78]. However, this process is slow and may take several hundred years to materialize [25], particularly for this long-lived and slow-growing conifer species. Our results also showed that the predicted current and future population tree height growth varied at large scales, and it was not conducive for among-population exchange of preadapted alleles and the enhancement of evolutionary adaptation [6]. Scientific evidence also suggests that natural migrations of forest trees are expected to lag climate change [9–12]. Thus, neither evolution nor migration are likely to provide a solution for *P. orientalis* to avoid the projected maladaptation in this and previous studies [18,19]. Therefore, assisted migration (moving the species to new habitats) and assisted gene flow (to move populations within their original habitats) using plantations appears to be a potentially effective approach to avoiding *P. orientalis* maladaptation under a rapidly changing climate. As an important ecosystem tree species, *P. orientalis* is commonly used for ecological restoration in arid mountain landscapes of northern China, so assisted migration is particularly important for afforestation and reforestation in China. Additionally, assisted migration has already played an important role in facilitating the adaptation of forest species and buffering the effects of climate change [79–81].

*4.5. Forest Management Practices to Promote Climate Change Adaptation*

In this study, the areas with suitable habitat of *P. orientalis* were predicted to increase in future climates using either local or optimal provenances. However, the growth potential would decline if local provenances were used, suggesting possible maladaptation. This finding also applies to the existing populations as they are local as well. On the other hand, when optimal provenances were used, improved growth would also be achieved along with the expansion of climatically suitable areas. There would be a huge potential to improve height growth by using the optimal provenances compared to using local counterparts, particularly in the barren lands of northern and northwestern China, where the application of optimal provenances could greatly enhance the productivity of *P. orientalis*, and increase the total suitable area. (Figure 6). Interestingly, because the prediction of optimal provenances is based on the climates of plantations and provenances, if similar climates were forecasted in different area of China, there will be multiple choices of optimal provenances.

For populations and regions at low risk of maladaptation, such as the local *P. orientalis* population in central China, silvicultural strategies should aim to enhance regeneration such that natural selection can continuously act on large numbers of juvenile trees [82,83]. For populations and regions at high risk of maladaptation to future climates, such as local *P. orientalis* population in northern and northwestern China, which is dominated by arid and semi-arid climates, our study shows that using optimal provenances is useful to avoiding maladaptation and to improve growth (Figure 5).

Nevertheless, these practical measures still need to be carefully evaluated in relation to current forest practices. First, despite the benefit of assisted migration using our model predictions, risks of transferring populations to new planting locations should also be considered [84]. The risks may include threats from new pests, pathogens, and hybridization with local populations that may dampen the advantage of the introduced optimal populations. In addition, there are other factors affecting tree performance beyond the consideration of our model predictions. For instance, extreme climate events may impose more severe impact on tree survival than climate normal that were used to make model predictions in this study, which are also used in the majority of other studies in this field [84]. However, extreme climate events are extremely difficult to predict; thus, to predict the impact of extreme climate conditions on tree performance remains a challenge. Enhanced atmospheric $CO_2$ may improve tree

water-use efficiency [85] and alleviate the maladaptation of local populations to climate change to some extent, but no model prediction considering the $CO_2$ effect has been reported in this species. Similarly, soil condition is also an important factor affecting the performance of tree populations. However, the impact of these two factors is not provenance specific, thus, not likely to considerably affect the applications of our model predictions. As fine scale global soil databases are becoming available (github.com/ISRICWorldSoil/SoilGrids250m), soil condition can be used an additional layer in our model predictions for planning new plantations with the consideration of assisted migration [86]. After all, considering the predicted impact of climate change, management decisions should be taken immediately due to the long time required to implement new forest management strategies and to convert highly vulnerable forests to less susceptible ecosystems [87–89].

## 5. Conclusions

Understanding the local adaptation and forest tree population response to climate is critical for developing adaptive forest resource management strategies. In this study, we found that *P. orientalis* populations are better adapted to their local climate conditions in central distribution areas than those distributed in peripheral regions. Suitable areas for *P. orientalis* are projected to increase in future climates, and the growth potential would decline if local provenances are used, suggesting the risk of maladaptation to the future climate for existing populations. However, using optimal provenances through assisted migration would increase growth potential. This would be particularly effective in northern and northwestern China. Thus, the assisted migration of *P. orientalis* has the potential to curtail losses in tree growth, and preserve forest health and productivity, subsequently maintaining ecosystem services in the coming decades.

**Supplementary Materials:** The following are available online at http://www.mdpi.com/1999-4907/10/8/622/s1, Table S1: Steps of climate variable selection and model optimization.

**Author Contributions:** J.-F.M., T.W., Y.A.E.-K. and Y.L. conceived and designed the study; X.-G.H., K.-H.J., S.-S.Z., N.C.C., S.-Q.J. conducted the experiments; X.-G.H., J.-F.M., T.W., and Y.A.E.-K. wrote the manuscript.

**Funding:** This research was funded by the National Natural Science Foundation of China (NO. 31670664) and the Fundamental Research Funds for the Central Universities (NO. 2018BLCB08).

**Acknowledgments:** The support provided by China Scholarship Council (CSC) during a visit of Xian-Ge Hu to University of British and Columbia is acknowledged.

**Conflicts of Interest:** The authors declare no conflict of interest.

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
