# Peer review of "Local Adaptation and Response of Platycladus orientalis (L.) Franco Populations to Climate Change"

_forests, doi:10.3390/f10080622_

Round 1

Reviewer 1 Report

Comments on Xian-Ge Hu et al – Local adaptation and response of Platyclades orientalis

The paper develops a Universal Response Function (URF) based on 69 provenances grown at 19 sites.  Local provenances performed better in terms of height at seven years at most sites, though non-local provenances were better at northern and northwestern sites.

This is not the first URF application, but dealing with data from numerous trial sites (measured at different ages) presented some challenges that had to be overcome. Appropriate methods and analyses are used and results are well presented. The data support the conclusions. The URF allows provenance variation in height growth to be examined in a way that will be of great interest to Forests readers. The paper is generally well written, though the English text lacks fluency in some places – some suggested changes are provided below. 

Title – okay

Abstract – “can predict the potential growth response…from any climate and growing in any climate conditions” – Don’t you mean in China? P. orientalis has been widely grown around the world (see map at https://www.gbif.org/search?q=Platyclades%20orientalis).  

Keywords – okay

Introduction

It would be good to have a couple of sentences putting URFs into the broader context of climate change studies of forests. Previous climate change studies of both natural and plantation forests have been dominated by the use of species distribution models (SDMs).  The development of improved climatic interpolation in the mid-1980s made SDMs (and later URFs) practical (see Diversity & Distributions, 20, 1-9 – this also mentions the first SDM climate change studies in natural & plantation forests which were published in 1988 – I don’t think these are mentioned in any of the references you cite).  A 2018 paper in Austral Ecology (43, 852-860) has a graph showing year of publication for 124 studies of climate change in forests from 1988-2016 – the vast majority of these were SDMs (see Dyderski et al. 2018 for details). SDMs can be applied to almost any tree species, but treat species as single homogeneous entities and don’t predict growth rates.  It’s excellent that URFs allow provenance variations in growth in response to climate change to be examined.

Materials and methods

Figure 1 – It’s not clear from Figure 1 how representative the provenances tested are of the entire distribution.  Wikipedia talks about distribution of the species in northwestern China, but other sources show a more eastern distribution (https://www.conifers.org/cu/Platycladus.php).  Could a natural distribution outline be added to Figure 1 or perhaps a map of herbarium records for China added as an inset?

Table 1 – the meaning of ‘individual’ needs to be explained in the text. Does it mean that you had access to individual tree height data from the 75 tree blocks in some cases, but only summarized trial mean data in some other cases?  If you had individual tree measurements for site 7 from ages  1-7 why couldn’t you calculate the trial mean?  23 for site 21 should presumably be 2-3.  

Results

Figure 4 – excellent illustration of the general response

Table 5 – what is the HT7 cut-off for considering an area as ‘suitable’?

Figure 6 – it’s a bit odd having the height growth scale increase from right to left. Why not use the same scale orientation as Figures 4 and 5?

 Discussion

Not only climatic conditions will change, but also atmospheric.  There should be some mention of likely impacts of rising CO2 levels on growth.  Has anyone developed a process-based model for P. orientalis?  I assume there might be some slight changes in absolute height growth, but relative growth rates between provenances would stay the same.

An obvious opportunity for future research might be to examine not only climatic, but also soil impacts on growth. Fine scale global soil databases are becoming available (see subsection 2.3 in For Ecol Manage, 430, 196-203).      

L. 413- 415 How practical is it to suggest widespread assisted migration for a slow growing species (“mean HT7 of about 2.0 m”) such as P. orientalis?

References

It’s good that you have picked up on early provenance climate change work such as that of Matyas (ref 16).  You may like to check out Ikeda et al (2017) Global Change Biology, 23, 164-176.  It’s interesting as it uses common garden data to improve ecological niche models, which are closely related to SDMs.

Minor suggested changes

L. 42 …and are projected…

L. 48 ..the vulnerability of spruce and beech would increase in the period to 2100…

L. 54 Delete ‘provenance’ and insert ‘provenances’

L. 64 applied to

L. 66 terms

L. 68 Delete ‘Thus’ and insert ‘Accordingly’

L. 70 Full species name needed at first occurrence in main text?

L. 73 Delete ‘in’ and insert ‘on’, newly established plantations

L. 76 impacts

L. 82 provide a scientific basis

L. 107 the response variable

L. 109 Delete ‘is’ and insert ‘were’

L. 174 Delete ‘current’ and insert ‘current climatic conditions’

L. 250 Delete ‘utilize’ and insert ‘show how’

L. 305 slowly

L. 324 – Delete sentence “We also predicted…at each planting site”. 

L. 326 Delete ‘improved’ and insert ‘substantially improved’, delete ‘also’

L. 345 The model built for Douglas-fir, based on observations in Austria and part of Germany, can predict…

L. 358 …over time to increase the frequency of genes relevant to local adaptability are largely understood

L. 360 …spatial variation among populations…

L. 369 …the performance of local provenances…

L. 407 species

L. 409 delete ‘the’, delete ‘enhances’ and insert ‘the enhancement of’

L. 410 Scientific evidence also suggests… forest trees…   

L. 411 Delete ‘likely’ and insert ‘likely to provide’

L. 414 …using plantations appears to be a potentially effective approach…

L. 426 …to improve height growth by using optimal provenances…

Reviewer 2 Report

Comments

In this manuscript, Hu et al. report findings of a modeling study aimed at understanding the response of Platyclades orientalis populations to climate change. They find that changes in climate could facilitate increased range of P. orientalis, but that local populations may be maladapted to these new conditions, even if range has expanded. However, performance was maintained when considering other, potentially more suitable populations instead of only local ones. The authors conclude that this suggests assisted migration may be a valuable tool in mitigating climate impacts on forests.  Overall, the study is valuable and of interest. It is also well designed and reported. There are a few main issues that should be addressed before the manuscript can be published.

Most importantly, it is not always clear that the statistics are structured appropriately. In particular, the vagueness with which parameters are described and the apparent lack of nesting of population within site are major issues. More detail is needed.

Throughout the manuscript, inconsistencies in labels make it challenging to follow the authors’ train of thought. Care must be taken to make sure that dependent and independent variables are clearly described and that references to each are consistent throughout the manuscript.

Finally, assisted migration is a controversial subject. Care must be taken when addressing it. As is, I don’t think the discussion explores the possible consequences or logistics sufficiently. Of course, this is not a review paper on assisted migration, but the subject deserves more attention than it is currently receiving.

Specific comments

70. provide genus

85. Objective 4 makes sense, but it not well established by the introduction

200. How did you determine which variables to use?

201. provide list of tested models and AICs in supplements

233. This is something that belongs in the figure legend, not necessarily the text.

236. Isn’t this expected based on the formulation of your allometric model?

244. Isn’t this redundant with above?

250. unclear

251. Population should be nested within site – it is not clear that this was done.

252. This is inconsistent with line 200.

257. Although means are important indicators of overall climate suitability, it is often the extreme events that have the largest impact on plant performance and ecosystem function. Therefore, variability in temperature and precipitation can be equally or more important in determining future suitability compared to mean values and is expected to change dramatically. This should be addressed somewhere in the manuscript.

293. Define local v optimal provenance

302. unclear

310. Be clear that you also ran these models for 8.5

330. Considering the majority of sites did not have observed HT7, and the apparent accuracy of the allometric relationship, could the data set be easily expanded to include more sites?

436. There should be some discussion of practicality of this. What are the risks associated with moving populations (e.g. could the species itself become weedy or are there pests/pathogens that could be moved alongside propagules to northern areas?). How hybridization between provenances dampen the effect?

436. How might competition affect these outcomes? If the primary evidence for maladaptation is lower growth, then performance of competing species is important.

438. Seed transfer guidelines?

Fig. 6. Line and text are too small

Reviewer 3 Report

The article is very interesting and has potential for publication. Yet, some methodological issues should, in my view, be addressed, namely:

Issue 1) As the trials come from different sources, it is important to understand if they were performed in the same years/seasons and/or if the same cultivation practices were applied among different sites. Authors should give information on this. Yearly climate data could be used to correct this, in case it is needed.

Issue 2) As only a few years are analysed it is also very important to understand if, in each site, the local provenances were tested in the same years/seasons as the other provenances. Yearly climate data could be used to correct this, in case it is needed.

Issue 3) Although it seem appropriate, authors could give some justification on the models used (both the linear models for the tree height estimation and the linear mixed-effects models).

Issue 4) Variable selection must be careful. Most of the referred variables seem quite evident to include.

Issue 5) Multicolinearity must be studied and the respective results presented.

Issue 6) It might be important to limit the predictions to the range of the observed maximum and minimum values, as these kind of models can produce quite erroneous extrapolations. For instance, the fact that the predicted areas increased towards arid and semiarid climates might not be spurious, given that CMD enters the model linearly (without a quadratic term).

Issue 7) It is important to check if the variable combinations obtained for the theoretical maxima (using the URF first derivate) do exist in the real world. An environmental distance could be used to check that. If the combinations are too far from a real-world existence, than the result might not be of any use.

Issue 8) The results are particularly relevant for forestry as height is a relevant trait. However, that is not so linear in terms of natural ecosystems as other variables relevant variables are not evaluated. Any reference to assisted migration should be avoided or eliminated.

Issue 9) In such a big territory, it might happen that similar climates occur far apart, containing different gene pools, thus different provenances. This could bring confusion to the URF and should be discussed by the authors.

Additionally, I send the following comments and suggestions:

Title: Platyclades should be changed to Platycladus. This change should be made in the entire document as this is the correct spelling of the Latin name.

Chapter 2.2 The estimation method is hard to understand and the description could be improved. In particular, it is advisable to keep the terminology rigidly fixed among the description (e.g. “general mean” is used both to refer to the “general mean over all sites” but also to introduce the “observed site mean height”). Although this is understandable it hinders the fluid comprehension of the method.

Discussion repeats some small parts of the previous texts (introduction, methods etc.) that could be eliminated or improved to focus more directly on the discussion.

Page 4, figure 1 caption: “trails” should read “trials” (?).

Page 6, line 184: Improve sentence: “A linear mixed- effects model (…) was used the following equation:”.

Page 7, line 212: Not sure what is meant with “become constants”. If possible, rephrase. They are still variables, not constants, they become duplicated in the model as X1i=X2i.

Page 7, line 214: Possibly you meant “spatial”, not “spatially”. Please confirm.

Page 7, line 218: How was the suitable area obtained? It is not clear what was done to calculate the percent change in area. Please clarify.

Page 9, lines 250-251: The first sentence should be reviewed, as it is not perfectly clear.

Page 13, lines 368-379: Worse performances on the limits of species distributions could also be expected directly from Hutchinson’s niche theory, in case they correspond also to suboptimal areas of the species own niche.

Page 14, line 407. “specie” should read “species”.

Round 2

Reviewer 2 Report

This manuscript is much improved over its previous version. I only have a few minor comments. 

Specific comments

42. move info on China to l.75+ when describing P. orientalis.

59. could not

213. reference supplement

467+. citations needed throughout this paragraph

Fig 3. It is very difficult to discern the trendlines, and the numbers to the right appear messy (also true for the way they are listed below).

Fig 6. Provide units for legend. It would also be helpful to see a map of the change in suitable habitats caused by selecting optimal v local to highlight areas most at risk. This would aid in some of the points made in your discussion.

Reviewer 3 Report

The article improved in clarity in many aspects. Authors answered satisfactorily for some of the raised issues. Please consider a thorough grammar checking for any new modification to the manuscript.

Nevertheless, some methodological issues were not appropriately answered and ought to be addressed:

Issue 3) In the first review I meant some justification for the use of such models a priori. I.e. in the methods section, authors could give some justification on the selection of such models (answering the question: why are such models appropriate to study the phenomenon?). In part, the newly added text contributes to such justification, but can be augmented and passed to the methods section.

Issue 4 and 5) In the first review I meant, also, a priori variable selection, i.e. before the modelling procedures. This must be careful in order to avoid a “shotgun approach” (select a large number of variables, independently of expected effects and thus without strong underpinning hypothesis). After this first selection, multicollinearity must be studied (e.g. using variance inflation factor). This is very important given the selected model types and given that significance values are used.

Cohen’s kappa might be used for pairwise comparisons, but multicollinearity with several variables must be addressed differently, at least using variance inflation factor.

Issue 6.1) In the first review I meant the maximum and minimum values of the independent variables (not the dependent variable). If a certain value of temperature or precipitation was not observed in the trials and is outside the range of the observed values, model output for such temperature/precipitation must be used with careful and eventually not used if too distant from the observed maximum and minimum. This should be addressed.

Issue 6.2) Concerning the new added text (lines 236-238), how do you limit the predictive range of the output? Do you discharge predictions outside the range? 

Additionally, I send the following comments and suggestions:

Page 2, line 75: “Platycladus. orientalis” should be changed to “Platycladus orientalis (L.) Franco” (delete the dot and include taxon authority).

Page 4, line 128 You still use “general mean:” to introduce the “observed site mean height”.

Page 4, figure 1 caption still reads “trails”. I believe it should read “trials” (?).

Page 9, line 269: The sentence “which showed that there are significantly variation exists” must be improved: “which showed that significant variation exists” OR “which showed that there was significant variation” OR “which showed, significantly, that variation exists” OR other?

Page 9, line 270: When referring to Table S1, please indicate which step you refer to (review this in the whole document).

Page 9, lines 269-271: “stepwise selection method. The best model (AIC = -543.88) were chosen that includes” maybe should read “stepwise selection method, the best model (AIC = -543.88) was chosen, which includes”.

Page 9, line 277. Sentence seems unfinished.

Page 14, line 421. “found” should read “findings”; “consistency” should read “consistent”; “environment” should probably read “environmental”; Please consider a thorough grammar checking for any new modification to the manuscript.
